# Replication Study of "Generative Causal Explanations of Black-Box classifiers"

## Reproducibility Summary

**Scope of Reproducibility**

We verify the outcome of the methodology proposed in the article, which attempts to provide post-hoc causal explanations for black-box classifiers through causal reference. This is achieved by replicating the code step by step, according to the descriptions in the paper. All the claims in the paper have been examined, and we provide additional metric to evaluate the portability, expressive power, algorithmic complexity and the data fidelity of their framework. We have further extended their analyses to consider all benchmark datasets used, confirming results.

**Methodology**

We use the same architecture and (hyper)parameters for replication. However, the code has a different structure and we provide a more efficient implementation for the measure of information flow. In addition, Algorithm 1 in the original paper is not implemented in their repository, so we have also implemented Algorithm 1 ourselves and further extend their framework to another domain (text data), although unsuccessfully. Furthermore, we make a detailed table in the paper to show the time used to produce the results for different experiments reproduced. All models were trained on Nvidia GeForce GTX 1080 GPUs provided by Surfsara's LISA cluster computing service at university of Amsterdam[1].

**Results**

We reproduced the framework in the original paper and verified the main claims made by the authors in the original paper. However, the GCE model in extension study did not manage to separate causal factors and non-causal factors for a text classifier due to the complexity of fine-tuning the model.

**What was easy**

The original paper comes with extensive appendices, many of which contain crucial details for implementation and understanding of the intended function. The authors provide code for most of the experiments presented in the paper. Although at the beginning their code repository was not functional, we use it as a reference to re-implement our code. The author also updated their code two weeks after we start our own implementation, which made it easy for us to verify the correctness of our re-implementation.

**What was difficult**

The codebase the authors provided was initially unusable, with missing or renamed imports, hardcoded filepaths and an all-around convoluted structure. Additionally, the description of Algorithm 1 is quite vague and no implementation of it was given. Beyond this, computational expense was a serious issue, given the need for inefficient training steps, and re-iterating training several times for hyperparameter search.

**Communication with original authors**

This reproducibility study is part of a course on fairness, accountability, confidentiality and transparency in AI. Since it is a course project where we interacted with other group in the forum, and another group also working with this paper has reached out to the authors about problems with the initial repository, we did not find necessary to do it again.

---

[1]This is our course project for the master course Fairness, Accountability, Confidentiality and Transparency in AI at the University of Amsterdam. Lisa cluster: `https://userinfo.surfsara.nl/systems/lisa`.

# 1   INTRODUCTION

Machine learning is increasingly used in different applications. The wide-scale spread of these methods places more emphasis on transparent algorithmic decision making, which has the potential to mitigate the potential for disruptive social effects. Yet, despite reliable results of complex black boxes, their internal reasoning and inner workings are not necessarily apparent to end-users or even designers. As a result, not even trained experts can grasp the reasoning behind forecasts. Moreover, modern legislation have necessitated the opportunity challenging these systems, especially in heavily regulated domains, increasing the need for machine learning systems that are (post-hoc) interpretable. [1].

Black-box artificial intelligence approaches like Deep Neural Networks have often proven to be able to capture complex dependencies within data, and allow for making accurate predictions. However, the actual internal logic used by systems dependent on such approaches is often nebulous or totally unclear. In this paper, we will reproduce the paper which focuses on the explainability aspect of AI.

Explainable Artificial Intelligence (XAI) refers to systems that seek to clarify how a black-box AI model achieves its performance. Post-hoc XAI achieves the desired explainability be generating reasons for decisions after having trained a black-box classifier. This is often achieved by extracting correlations between input features and the eventual forecasts[2]. This paper aims to reproduce such a post-hoc XAI algorithm, capable of providing clear and interpretable explanations for complex black-box classifiers [3]. The central contribution made by [3] is placing explanations in a causal-generative framework. By using a variational auto-encoder (VAE) [4] , low dimensional representations of the data can be achieved. By further incorporating a mutual information loss between the classifier and the latent variables, the latent space is decorrelated into factors that actively influence the classifier's prediction, and those that capture superfluous variation in the underlying dataset. The causal-VAE , dubbed a generative causal explainer (GCE) by [3] can be studied and intervened with to provide a powerful framework for inducing explainability.

There exists literature suggesting that some studies cannot be replicated [5]. This is valid also for the most respected journals and conferences in AI. Steps must be taken to ensure high trustworthiness of AI algorithms [6]. In the experiments presented here, we re-implement the framework used by [3], extend their analyses to all benchmark datasets discussed and provide an extension to a novel domain.

When beginning our process, the officially provided code repository corresponding to the authors' publication was not capable of running in all discussed scenarios, nor producing results similar to those presented [2]. However, our re-implementation has ascertained all results, and been extended to incorporate all analyses, and can report their framework to be reproducible.

# 2   Scope of Reproducibility

The original paper establishes a causal inference framework. While intuitive, this introduces the additional challenge of balancing causal intervention with expressive latent variables. To overcome this hurdle, [3] integrate causal inference into the VAE learning objective. This system consists of two basic components: a way to describe the data distribution, and a method of generating a consistent classification model. In order to get a clear objective of the reproduction, we consider the the main claim(s) of the original paper as:

- Claim 1: creating a generative framework that describes a black-box classifier, and a process that achieves this while maintaining latent descriptions that result in high data fidelity and scalability of the overall system

- Claim 2: their objective function helps the generative model disentangle the latent variables into causal and non-causal factors for the classifier. Their approach is sufficient for any black-box classifier that can provide class gradients and probabilities with respect to the classifier input

- Claim 3: the resulting explanations show what is important to the accompanying classifier

- Claim 4: the learned representation may also be used to explain counterfactual explanations as it incorporates both generative and causal modelling

The standard of reproducibility demands that machine learning methods be routinely evaluated on the verifiability of their results. The following additional metrics (properties) will be used as measurement of reproducibility:

---

[2]Since, the authors have responded to suggestions and rewritten their code base substantially.

- Portability: the modelling domains and types of machine learning models that can benefit from their framework

- Expressive Power: the explanation structure that can be produced by their framework

- Algorithmic Complexity: the computational complexity of their algorithm

- Data Fidelity: the degree of precision of the data will range from low to high fidelity. In cases where high-faithfulness data to train the model are not necessary, low-faithfulness data often may be used. The small amount of usable data will greatly influence the model's ability to yield accurate estimates.

## 3  PROJECT DESCRIPTION

As stated, we evaluate the performance of the method proposed in paper[3] as a causal post-hoc explainer of black-box classifiers by reproducing the code necessary. At the beginning of this project, the code provided by the authors was not in a usable state. Hence, we decided to reimplement the code for the necessary architectures and conduct experiments with the implementation details provided in the paper. By the time we finished our implementation, the project repository had been updated to be able to reproduce the initial claims in the paper.

Beyond just re-implementing already existing code, two extensions were considered. First, the paper suggests a technique for selecting the $K$, $L$ and $\lambda$ hyperparameters used to train the generative model (Algorithm 1). However, no implementation of this algorithm is present in the authors' code. To test its validity and because of the tedious and time-consuming nature of manual hyper-parameter search, we implemented the automated algorithm using reasonable assumptions.

Second, in an effort to verify the robustness of this method, similar experiments for image classification were also conducted using a textual domain. Compared to the simple image benchmark datasets used in the initial experiments, text classification and generation are considerably more complex. These models will be used to test the scalability of the proposed framework.

## 4  METHODOLOGY

The Generative Causal Explainer (GCE), is at its core a generative model with awareness of a discriminative black-box classifier. For the generative model, this paper exclusively uses the Variational Auto-Encoder (VAE) [4]. While the VAE allows expression of a dataset in terms of a low-dimensional posterior distribution, the addition of the classifier allows for disentangling the proposed latent space into variable subsets that causally influence decisions of the classifier and those that capture superfluous variance. The former subset is denoted $\alpha$, whereas the latter is denoted $\beta$, having cardinalities $K$ and $L$ respectively.

The goal of this modelling framework is to learn a generative mapping $g : (\alpha, \beta) \to X$ that further satisfies the following criteria: $p(g(\alpha, \beta)) \approx p(X)$, the factors $(\alpha, \beta)$ are statistically independent and $\alpha$ has strong clausal influence on the classifier's output $Y$. The proposed objective function of this framework is thus,

$$\underset{g \in G}{argmax} \quad \mathcal{C}(\alpha, Y) + \lambda \cdot \mathcal{D}(p(g(\alpha, \beta)), p(X)) \tag{1}$$

where $g$ is the GCE model that satisfies the constraints from the set of possible generative models $G$, $\mathcal{C}(\alpha, Y)$ is a metric that quantifies the causal influence of $\alpha$ on $Y$ and $\mathcal{D}$ is a variational lower bound that measures the proximity of $p(g(\alpha, \beta))$ to $p(X)$. The inclusion of the $\mathcal{D}(p(g(\alpha, \beta)), p(X))$ is necessary to ensure the generated explanations remain in, and ideally closely approximate, the data distribution.

While there are several candidates for a causal influence metric, the original authors opted for the information theoretic motivated mutual information (MI). O'Shaughnessy et al. offer several reasons for choosing MI, including its compatibility with deep neural networks, its ability to quantify indirect causal links between the GCE's latent space and the classifier, and its equivalence to 'information flow' in the proposed causal model when considering do-calculus. Thus, in the above provided loss function,

$$\mathcal{C}(\alpha, Y) = I(\alpha; Y) = \mathbb{E}_{\alpha, Y} \left[ \log \frac{p(\alpha, Y)}{p(\alpha)p(Y)} \right], \tag{2}$$

where $I(\alpha; Y)$ is the aforementioned MI between the causal factors and the classifier. No closed form solution computation for Eq. 2 is provided. Rather, a Monte-Carlo estimator is employed, using data samples drawn from the GCE posterior and their classifications by the accompanying black-box classifier. For detailed explanation of the underlying method, we direct the reader to Appendix D of [3]. Note that this estimation method requires passing

drawn samples through both the GCE's decoder network and the classifier, and for low variance estimates of $I(\alpha; Y)$, numerous estimates are required. As such, estimating this quantity at every training step is computationally expensive, especially when considering the cost of a vanilla VAE.

### 4.1 GCE Architectures

As VAEs do not explicitly limit the architectures used in the encoder and decoder networks, much like the classifiers in question, they can make use of the same inductive knowledge encoded into the black-box classifiers. Thus, for image classification, given the performance of convolutional neural networks, a similar set of networks can be considered for the GCE. Naturally, while replicating and unspecified, the same architectures as used by O'Shaughnessy et al. are used here as well.

The image classification datasets used (see Sec. 5) are benchmark datasets of low complexity. Hence, both the classification and generative models were limited to shallow neural networks. The classifiers consisted of 2 ReLU activated convolutional layers fed into a max-pooling layer before 2 ReLU activated linear classification layers. Drop-out was present prior to either linear layer. In all instances, this sufficed for achieving near perfect accuracy. Both the encoder and decoder used 3 layers of convolution (transposed for the decoder), with additional linear layers for converting the feature maps. Specifics for the models used are given in Table 3.

For text classification, the core architecture used was the Long Short-Term Memory (LSTM) network [7]. The classifier consisted of a bi-directional LSTM, whose hidden states at every time-step were concatenated and fed into a 3 successive convolution/ReLU/max-pooling blocks, before being projected into classification nodes via a linear layer. The embeddings used came from the 840b token Common Crawl pre-trained 300 dimensional GloVe vectors, limited to the vocabulary present in the SST dataset[3]. Ultimately, this model achieves 84% accuracy on the binary SST classification task. The encoder and decoder nets used a common VAE generation architecture [4], consisting of single layer LSTM, with embeddings not using pre-initialised weights. The hidden and cell states at the last time-step were used for predicting the input-dependent posterior statistics, with the posterior samples being fed into the decoder as the initial hidden and cell states, while also being appended to every embedding vector.

Posterior collapse, a situation where the decoder network essentially ignores the encoder output, proved a serious problem for text generation. To overcome this issue, an aggressive training regiment was used [8]. Here, the encoder network is trained until convergence before updating the decoder network, resulting in stable and informative signals for the decoder network. This method of training is necessary for the first few epochs, but quickly ameliorates the situation and allows for regular VAE training to continue. However, given the sheer computational expense of using aggressive training, combining this with MC-estimation of $I(\alpha; Y)$ would quickly prove intractable. As such, both the classifier and generative model were first trained disjoint, resulting in a regular VAE text-generator, before attempting to fine-tune into a functioning GCE.

### 4.2 Implementation

Upon start of this project, the original authors' official code repository was non-functioning. Hence, all models were implemented from scratch using the descriptions and guidelines provided in the original paper. The only exception to this was the MC-estimation for information flow, although this was further optimised during the project. We exclusively used PyTorch, PyTorch-Text and PyTorch-Lightning for implementation, and all models were trained on Nvidia GeForce GTX 1080 GPUs provided by Surfsara's LISA cluster computing service.

## 5 EXPERIMENTAL SETUP AND CODE

### 5.1 Datasets

Experiments using image classifiers are conducted on the traditional MNIST hand-written digits [9] and the newer Fashion MNIST (fMNIST) datasets [10]. The official training set of traditional MNIST was split into training and validation subsets of 50,000 and 10,000 images, respectively. The test set remained the same as the original dataset, composed of 10,000 images. Only the images labelled '3' or '8' were used to train the binary 3/8 classifier, whereas images labelled '1', '4' and '9' were selected to train the 1/4/9 classifier.

For fMNIST, the training set remains the same as the original dataset containing 60,000 images. The test set is divided into a validation set and a test set, containing 6,000 and 4,000 images, respectively. The t-shirt, dress, and coat images, labelled '0','3', and '4', were used to train the 0/3/4 classifier. Both traditional MNIST and fMNIST were limited to

---

[3]Available at https://gist.githubusercontent.com/bastings/b094de2813da58056a05e8e7950d4ad1/raw/3fbd3976199c2b88de2ae62afc0ecc6f15e6f7ce/gl

samples with the labels of interest. All images were scaled to size $28 \times 28$. In both datasets, the train/validation/test splits was done using the file indices.

Experiments conducted using text classification used the Stanford sentiment treebank [11] movie reviews corpus. Here the officially recommended train/validation/test splits were used. Rather than using the 5-class fine-grained classification, only positive and negative reviews were used, with the 'very-' classes being converted to their less polar alternatives.

### 5.2 Hyperparameters

In the reproduction experiments, hyperparameters are set to be the same as the original paper. The lists of hyperparameters of CNN classifier and GCE model can be found in Table 4 and Table 5, see Appendix A.

Beyond just the values provided, however, Algorithm 1 from the original paper was also implemented to conduct a hyperparameter search for $K$, $L$ and $\lambda$. This procedure was not rigorously defined in the original paper, using terms that are left open for interpretation, such as "plateaus" or "approaches". Due to this, some assumptions were made in the process:

- "Plateauing" was defined as the value in question achieving a local optimum, with the next iteration reversing its trend.
- "Approaching the value from step 1" was defined as either coming within a certain percentage threshold of the target value, or being closer to the target value than the next iteration.
- $\mathcal{D}$ and $\mathcal{C}$ were defined as loss values subject to minimization.

This technique requires three parameters to be chosen:

1. $\xi$: a factor that dictates how close to the $\mathcal{D}$ obtained in step 1 a value must be to be considered as having approached $\mathcal{D}$.
2. $\lambda_0$: the value of $\lambda$ to start with.
3. $\kappa$: the factor by which to increase $\lambda$.

In our hyperparameter search experiments, we use $\xi = 0.05$, $\lambda_0 = 10^{-3}$ and $\kappa = 10^{0.5}$.

### 5.3 Model Training

In reproduction experiments, we trained a GCE model to generate explanation factors for image classifier outputs. The CNN classifier was trained for image recognition task using SGD as optimizer. The network architecture is shown in Table 2 and the hyperparamters settings are listed in Table 5. The GCE model is trained to maximize the objective 1 with Adam optimizer. We use the values of K, L and $\lambda$ suggested in the original paper. Hyperparameter details can be found in Table 4.

## 6 RESULTS

### 6.1 Results reproducing original paper

Using the GCE model described in Section 4.1, explanations for the CNN classifiers trained on MNIST and fMNIST datasets were found. The latent factors $\alpha$ and $\beta$ are visualized in Figure 1, showing exactly how $g(\alpha, \beta)$ and the classifier output change as the latent factors are modified for the 3-8 classifier. One can observe that $\alpha_1$ influences the features that separate the digits 3 and 8 (the classifier's output being given by the colour surrounding the digits) while retaining stylistic features unrelated to the classifier such as skew and thickness. By contrast, non-causal factors $\beta_i$ controls features irrelevant to classifier outputs. As shown in Figure 1(b-d), changing $\beta_i$ leads to stylistic changes of digits but does not affect classifier predictions.

By visualizing high-resolution latent factor sweeps in Figure 2, the GCE model can assist a practitioner in identifying important data features for classification results. As shown in the first row from the top in Figure2, the digits '4' smoothly transition into '9' by completing the loop of the digit '9' while the digit stem remains fixed. Finally, the '9' digit gradually transitions to a '1'.

To verify the causal influence of the GCE on the classifier, the information flow is studied, along with an ablation study of individual factors on classifier performance. Figure 3(a) shows the information flow from $\alpha$ factors to Y is high while

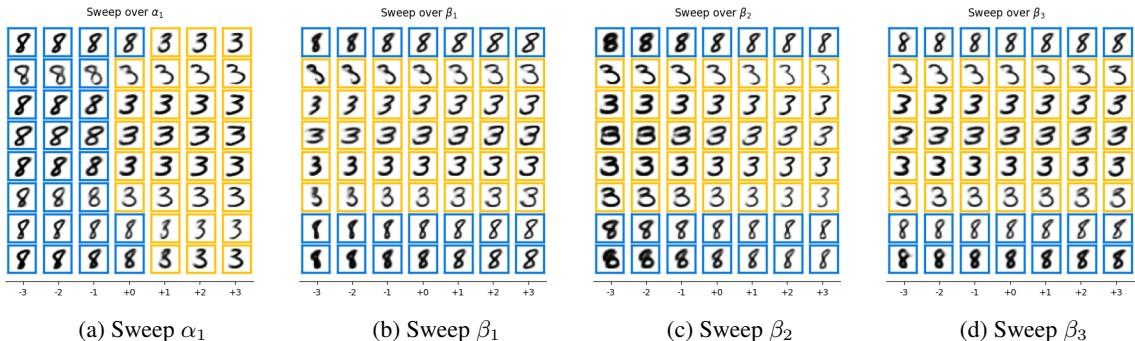

| (a) Sweep $\alpha_1$ | (b) Sweep $\beta_1$ | (c) Sweep $\beta_2$ | (d) Sweep $\beta_3$ |

Figure 1: Visualizations of learned latent factors for MNIST classifier trained on classes '3' and '8'. The colour denotes the classifier's decision, with each adhering to a specific class. The axis gives the additive value to each latent dimension. Complete results are in Figure 4 for MNIST 3-8, Figure 5 for MNIST 1-4-9, and Figure 6 for FMNIST 0-3-4.

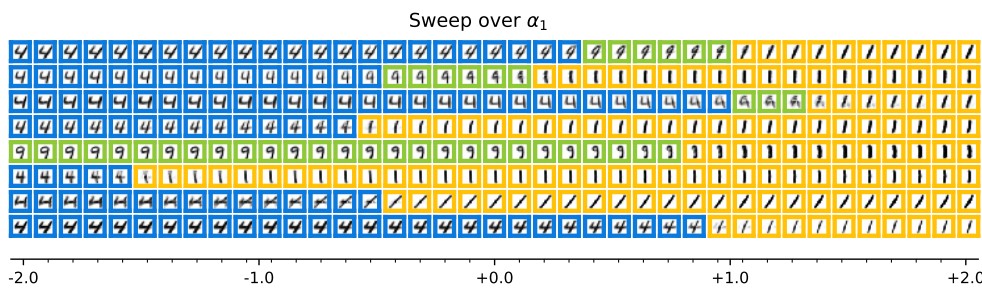

Figure 2: High-resolution transition regions of the first causal factor in explaining the MNIST 1/4/9 classifier.

the information flow from $\beta$ factors to Y is low. For the ablation study, we delete individual data points from the data by fixing individual latent factors in each validation data set to different random values taken from the prior $N(0,1)$. This decrease in the precision of the classification is seen in Figure 3 (b). Note that modifying aspects influenced by causal factors degrades the accuracy of the classifier substantially, whereas elimination of non-causal aspects only has a marginal effect on the accuracy of the classifier.

Our implementation reproduces the results in the original paper and supports the authors' claim that their method is able to separate causal and non-causal factors of a classifier. The learned latent factors can then be applied to explain classification decisions of a classifier.

## 6.2 Results beyond original paper

Figure 7 shows results from the hyperparameter search using the version of Algorithm 1 described above with the 3/8 classifier. This corresponds to Figure 11 in the original paper. Results for varying $\lambda$ are computed for all values of K but only shown for $K = 1$. The final parameter values selected by the procedure are $K = 1$, $L = 9$ and $\lambda = 0.001$. Results of using these parameters to train the GCE can be seen in Figure 8. While the parameters found here differ from the ones presented in the original paper, the difference in results obtained when they are used is not significant.

As a final extension, an attempt was made to using an altered GCE set-up on a text-domain. While the individual components of the GCE performed well (see Appendix C for some samples from the text VAE), fine-tuning the combination into a functioning explainer was not successful.

Table 1 shows the amount of time required to train each model. It is clear that training a vanilla VAE takes more computational power than training a classifier on the same data. However, an even larger cost is induced by the causal loss calculation required for the GCE; GCE models consistently took more than five times longer to train than their VAE counterparts with the same architecture.

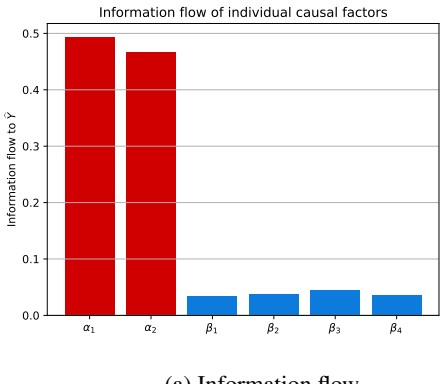
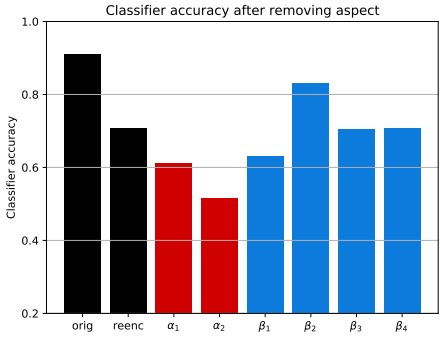

(a) Information flow.                    (b) Removal of latent factors.

Figure 3: Ablation study for training FMNIST dataset: (a) each latent factor affecting the classifier performance measured by information flow. (b) comparison of classifier accuracies when data aspects regulated by the individual latent factors are eliminated. (Figure 5ab in the original study). The figures here are generated using the fMNIST data-set.

Table 1: Training time of classifiers, GCE models and vanilla VAE models (GCE models trained without the causal loss term).

| Dataset | K+L | Duration (hrs) | | | Duration (relative to classifier) | | |
| | | Classifier | VAE | GCE | Classifier | VAE | GCE |
|---------|-----|------------|-----|-----|------------|-----|-----|
| MNIST | 3, 8 | 8 | 00:01:11 | 00:05:24 | 00:34:25 | 1.00 | 4.56 | 29.08 |
| MNIST | 1, 4, 9 | 4 | 00:02:47 | 00:05:17 | 00:25:04 | 1.00 | 1.90 | 9.01 |
| FMNIST | 0, 3, 4 | 6 | 00:02:59 | 00:03:58 | 00:23:40 | 1.00 | 1.33 | 7.93 |
| SST | Pos, Neg | 32 | 00:02:49 | 05:15:09 | - | 1.00 | 111.89 | - |

# 7    DISCUSSION

Given the results presented above, and their proximity to those presented in the original paper, we tentatively verify all claims presented in Sec. 2. The GCE model produces high-quality examples that seem to align with the classifier's internal decision-making process. Furthermore, by using interventions in the GCE's posterior, information regarding features important to the black-box classifier were made apparent. Such a framework also clearly supported the use of counterfactuals, with alterations in the causal factors seeing the class change, and the stylistic interpretation of the produced examples remaining unaltered. Lastly, the computation of information flow to individual factors and the performed ablation study (now extended to all initial experiment domains), clearly show the success of mutual information in disentangling the GCE's latent space into causal and non-causal factors.

However, upon implementing GCEs for simple classification models, the scalability of the proposed framework can already be drawn into question. As mentioned in 4.1, the introduction of a MC-estimated quantity like mutual information has significant impact on the computational expense required for training, essentially forcing significantly more passes of data through the decoder and classifier networks for a single weight update. Even for the relatively sparse CNN-based GCE and classifier, the suggested number of $\alpha$ and $\beta$ estimates in conjunction with current implementation, implied 2500 additional forward passes for a single backward pass. All our experiments indicated this being the bottleneck of the modelling pipeline. Future research into optimising this process, for example by ensuring lower variance estimates or mixing of new sample estimates with older generations, could prove valuable in extending the original research.

The issues with computational efficiency were strongly exacerbated by the requirement of a more complex generative model for the text domain. In fact, given the use of an alternative training regiment, incorporating information flow to induce causal disentanglement would have made training until convergence virtually intractable. While the eventual failure to fine-tune from a functioning language generation model to a GCE could be an artefact of the pathologies plaguing text generation using auto-regressive architectures, it also speaks to the potential of portability for this framework. Using a classifier to produce interpretable understanding of the latent space in such a language generation model could prove tremendously interesting, allowing for a causal framework similar to the work of [12]. Furthermore,

being able to fine-tune pre-trained VAEs into GCEs would provide the suggested framework far more flexibility, essentially addressing the computational efficiency issues mentioned before.

### 7.1 Shortcomings of the original paper

The greatest shortcoming found in the original paper is the failure to address the scalability problem of the approach, along with the lack of rigour when describing the hyperparameter selection technique.

### 7.2 REFLECTION: What was easy? What was difficult?

It was not trivial to re-implement the proposed method because the specifics and some details required for the implementation do not appear in the paper. However, we still managed to reproduce the results in their paper. In order to extend the algorithm to another domain, code modifications were required. No instruction is given in their original repository on how this could be done, which makes it difficult to extend this framework or apply it to other domains without reading their paper and code in depth. Having access to the (updated) codebase was quite helpful however, as it includes some implementation specifics that are not mentioned in their paper, which we made use of as a source of reference when implementing and debugging our own repository.

## 8   CONCLUSION

While some issues and discrepancies were encountered while re-implementing, ultimately we conclude that the original paper combined with the official repository are enough to validate the claims of [3]. Results were comparable, and indeed led to high-quality explanations. However, while the central idea is elegant and is now proven to work, we bring into doubt the extensibility of their approach. Due to the computational expense required, it is likely that the GCE models introduced will only function, in their current implementation, for small datasets and simple classifiers. Finally, this project confirms just how difficult it is to make implementations of AI transparent and reproducible.

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

# APPENDIX

## A. Neural network architectures

| Network architecture for CNN classifier |
|---|
| Input (28×28) |
| Conv2 (32 channels, 3×3 kernels, stride 1, pad 0) |
| ReLU |
| Conv2 (64 channels, 3×3 kernels, stride 1, pad 0) ReLU |
| MaxPool (2×2 kernel) |
| Dropout (p = 0.5) |
| Linear (128 units) |
| ReLU |
| Dropout (p = 0.5) |
| Linear (M units) Softmax |

Table 2: Network architecture for CNN classifier

| Encoder | Decoder |
|---|---|
| Input (28×28) | Input (K + L) |
| Conv2 (64 chan., 4×4 kernels, stride 2, pad 1) | Linear (3136 units) |
| ReLU | ReLU |
| Conv2 (64 chan., 4×4 kernels, stride 2, pad 1) | Conv2Transp (64 chan., 4×4 kernels, stride 1, pad 1) |
| ReLU | ReLU |
| Conv2 (64 chan., 4×4 kernels, stride 1, pad 0) | Conv2Transp (64 chan., 4×4 kernels, stride 2, pad 2) |
| ReLU | ReLU |
| Linear (K + L units for both $\mu$ and $\sigma$) | Conv2Transp (1 chan., 4×4 kernel, stride 2, pad 1) |
| | Sigmoid |

Table 3: GCE network architecture used for MNIST and fMNIST experiments

Table 4: Hyperparameter settings for GCE models

| Parameters | Dataset | | |
|---|---|---|---|
| | MNIST | MNIST | FMNIST |
| classes | 3,8 | 1,4,9 | 0,3,4 |
| K | 1 | 2 | 2 |
| L | 7 | 2 | 4 |
| $\lambda$ | 0.05 | 0.1 | 0.05 |
| training steps | 8000 | 8000 | 8000 |
| learning rate | $5 \times 10^{-4}$ | $5 \times 10^{-4}$ | $10^{-4}$ |
| $N\alpha$ | 100 | 75 | 100 |
| $N\beta$ | 25 | 25 | 25 |
| batch size | 64 | 64 | 32 |

Table 5: Hyperparameter settings for CNN classifier

| Parameters | Dataset | | |
|---|---|---|---|
| | MNIST | MNIST | FMNIST |
| classes | 3,8 | 1,4,9 | 0,3,4 |
| learning rate | 0.1 | 0.1 | 0.1 |
| momentum | 0.5 | 0.5 | 0.5 |
| batch size | 64 | 64 | 64 |
| epochs | 20 | 30 | 50 |

## B. Additional Reproduction results

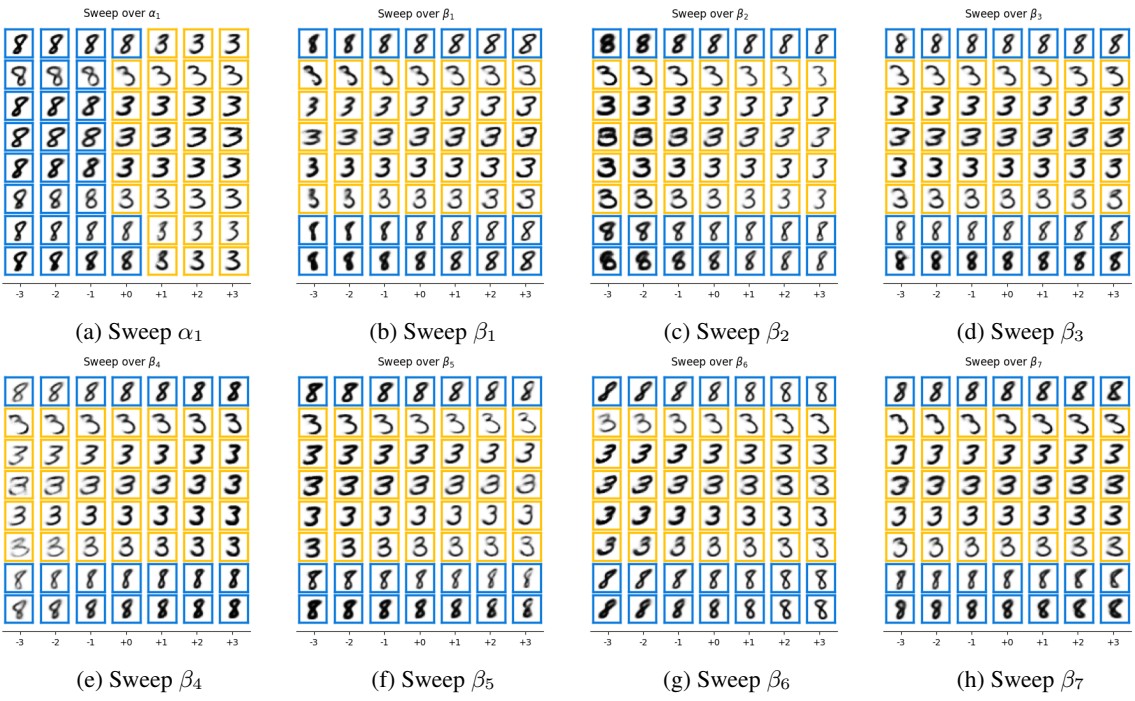

(a) Sweep $\alpha_1$  (b) Sweep $\beta_1$  (c) Sweep $\beta_2$  (d) Sweep $\beta_3$

(e) Sweep $\beta_4$  (f) Sweep $\beta_5$  (g) Sweep $\beta_6$  (h) Sweep $\beta_7$

Figure 4: Visualizations of learned latent factors for MNIST classifier trained on classes '3' & '8'

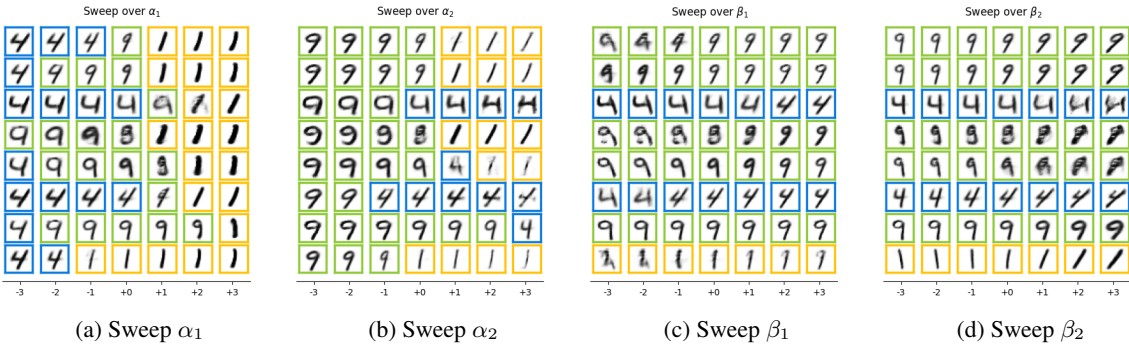

(a) Sweep $\alpha_1$  (b) Sweep $\alpha_2$  (c) Sweep $\beta_1$  (d) Sweep $\beta_2$

Figure 5: Visualizations of learned latent factors for MNIST classifier trained on classes '1' & '4' & '9'

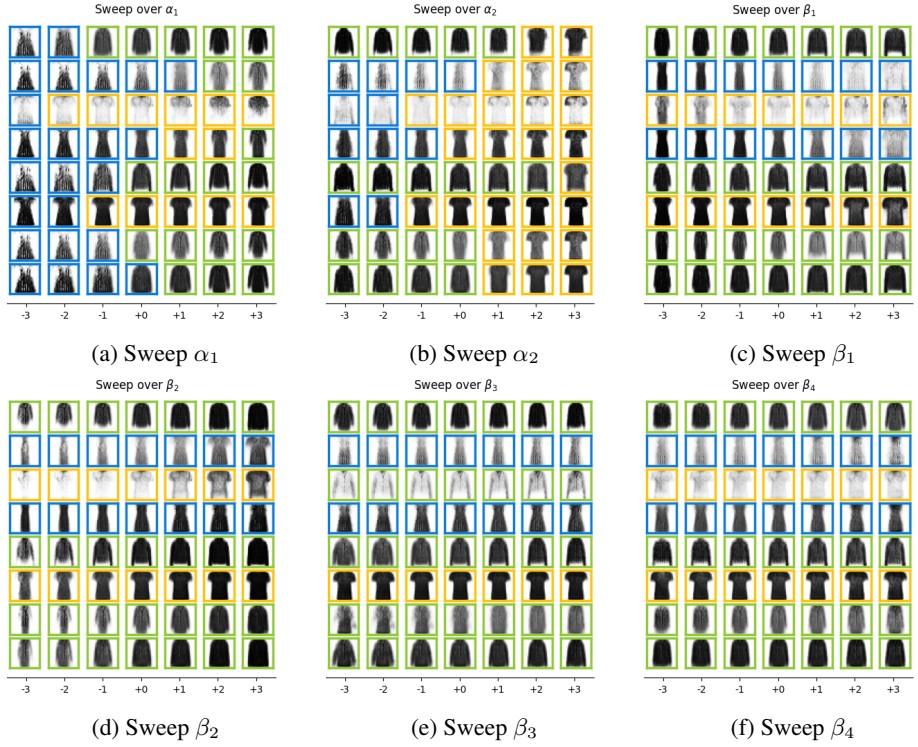

Figure 6: Visualizations of learned latent factors for fMNIST classifier trained on classes 't-shirt-top,' 'dress,' and 'coat.'

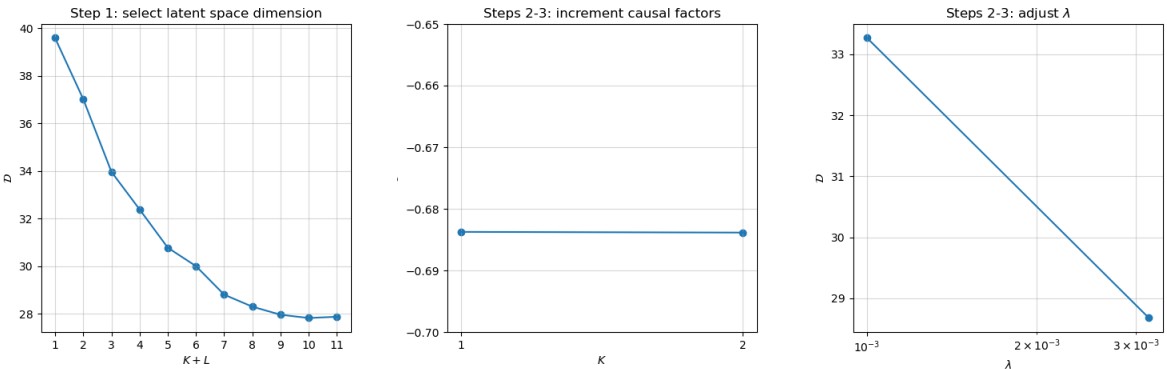

Figure 7: Results of parameter selection technique for the 3/8 classifier.

## C. Text Explainer

The text classifier used a bi-directional LSTM of 256 units. As mentioned the embeddings had been initialised using pre-trained GLoVe embeddings. All hidden states were then concatenated into a single feature map. Blocks of convolutional layers (32 filters of kernel size 3), ReLU activations and max-pooling (kernel size of 2 and stide of 2) were applied to reduce these feature map sizes. Ultimately all feature maps were projected down into the required number of classes using a linear layer. Adam was used as the optimizer, using an initial learning rate of 1e-3 and decaying this by a factor of 0.85 at every epoch.

The text-VAE follows the proposed structure given in [8] as closely as possible. The embeddings used 512 dimensions and the single layer LSTM 1024. The last hidden and cell states were projected down to 32 latent dimensions. Aggressive training was used until the mutual information between the latent variables and the encoder input stabilised (typically 5 epochs). For the inner loop in the aggressive training schedule, 250 iterations were allowed, after which

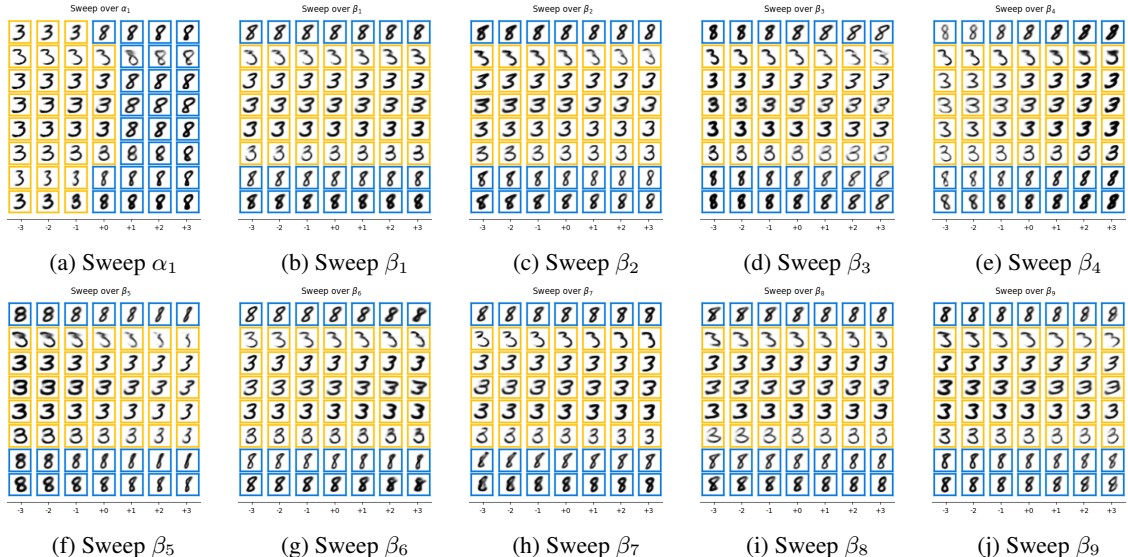

Figure 8: Visualizations of learned latent factors for MNIST classifier trained on classes '3' & '8' with the hyperparameters obtained from the hyperparameter search procedure.

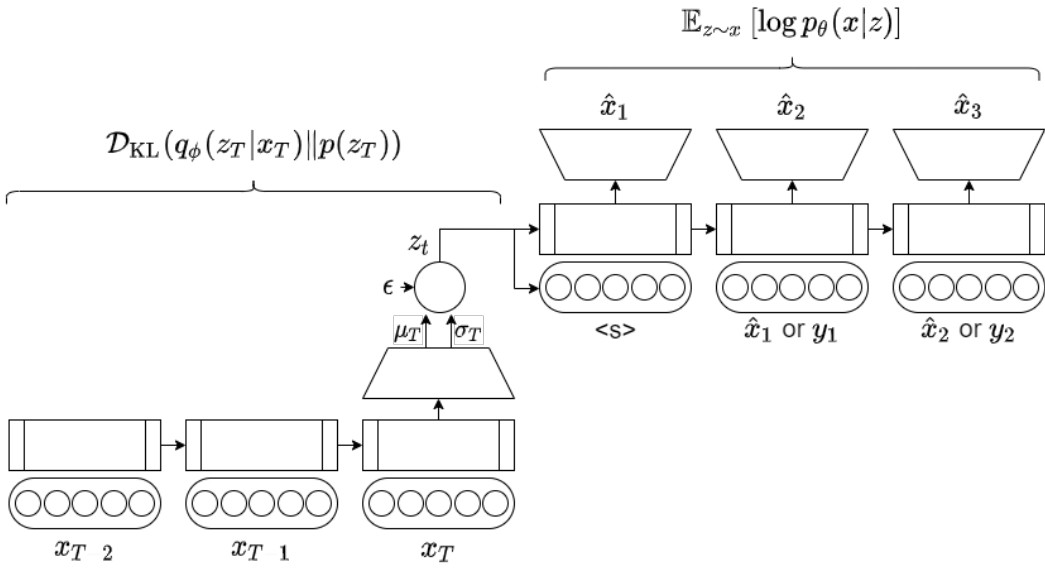

Figure 9: Text-VAE architecture.

it was assumed the encoder had converged. The Kulbback-Leibler divergence was weighted using a linear annealing schedule between the first 10 epochs.

Fine-tuning was attempted using $\lambda = 1 - 3$ and $lr = 1e - 3$ using the standard GCE training scheme. These hyperparameters empirically proved to induce lowest lowest causal loss after a gridsearch. Ultimately, no model improved it's causal loss after more than 3 epochs.

Table 6: Samples drawn from the Text-VAE and the GCE that was trained from it.

| Text-VAE | Text-GCE |
|---|---|
| plays like an unbalanced mixture of flavours and violence the movie unfolds with an undeniable mixture of dignity that sneaks up in the viewer | -2.00: but and heartwarming it does and involving and involving and involving and evocative visuals and |
| | -1.00: but and heartwarming it does and involving and involving and involving and involving |
| highlighted by a gritty cast but it makes you feel like a character but ends up as shallow and unsettling as one | 0.00: but and it works and ponderous and romantic flicks and clich and |
| | +1.00: but and it works and ponderous and snipes and romantic flicks and clich |
| at times a bit melodramatic it would be a bit better than you d expect if you ve never seen it all before | +2.00: but and heartwarming it does and clich and clich and clich |

