# OpenReview forum: "Replication Study of “Generative Causal Explanations of Black-Box classifiers”"
_ML_Reproducibility_Challenge/2020 — RC2020_

### Official Review · AnonReviewer2 · 2021-03-01
**Solid work but need more claims about the solution to reduce algorithm complexity**

**Rating:** 8
**Confidence:** 2

**Review:**

Thank you for your great paper!

The paper successfully proves that the original paper is reproducible and it could provide the post-hoc casual explanations for black-box classifiers through the casual reference. Furthermore, the paper establishes its own evaluation system to evaluate the original paper from different aspects. Moreover, the paper extends the application domain from images to texts, which is great for generalization. However, I think it would be good if you can add more details about what is different between your implementations and the original paper's, like the solution to reduce algorithm complexity. Last but not least, adding a few figures about the models' architecture would be great for readers!

**Familiar With The Original Paper:**

I have not read the original paper

**Reproducibility Summary:**

Report has summary

---

### Official Review · AnonReviewer3 · 2021-03-10
**Good Job in Reproducing the Paper.**

**Rating:** 9
**Confidence:** 4

**Review:**

Good job in reproducing the results. The original paper carries out methods for generating causal post-hoc explanations of black-box classifiers based on a low-dimensional representation of input data. This paper tries to reproduce those results in detail and provide a more efficient implementation. While reproducing the results of the original paper, the authors of this paper take a further step ahead:
1. They provide a higher resolution transition for the first causal factor for MNIST 1/4/9 classifier.
2. They also dive into hyperparameter search for the Principled procedure for selecting (K, L, λ) as explained by the original paper.
3. They have also tried out the proposed method on the SST text dataset and tabulated the duration for both text as well as for image datasets.

Also, the resulted figure from this implementation is similar to the figures reported in the original paper.

They have also mentioned that they have not found this method to be scalable in contrary to the original paper but they have not mentioned any ideas on how to scale up but they are relying on future papers to do so.

On a final note, this is a solid work and will be very helpful to gain insights if the original paper is reproducible or not and to what extent can the algorithms mentioned in the paper can be used to solve the explainability problem of black-box classifiers.



**Familiar With The Original Paper:**

I have read the original paper

**Reproducibility Summary:**

Report has summary

---

### Decision · Program_Chairs · 2021-03-31

**Decision:**

Accept

**Comment:**

Selected for ReScience-C Journal Publication.

Strong reviews. The paper also goes above and beyond by exploring other datasets.